# Cardiac-Device Implantation and Pneumothorax—A Symptom-Based Approach: Experience from a District General Hospital

## Grace George and Avinash Aujayeb *

Respiratory Medicine Department, Northumbria Healthcare NHS Foundation Trust, Cramlington NE22 9EH, UK
* Correspondence: avinash.aujayeb@nhct.nhs.uk

**Abstract:** In patients undergoing cardiac-device implantation, pneumothorax incidence occurs in 1–6%, and is more common in women over 80 years of age with chronic obstructive pulmonary disease (COPD). The aim of this study was to do a service review to identify ways to improve care delivery. Caldicott approval was gained. Those complicated by a pneumothorax were further analysed regarding basic demographics and pleural interventions and outcomes. Continuous variables are presented as mean (range) or median (with interquartile ranges) and categorical variables are presented as percentages where appropriate. A total of 2056 implantation episodes from January 2010 to December 2020 occurred with 70 pneumothoraxes (3.4%), which were all related to pacemaker insertion. The mean age was 68.1 years (17–97), 39 were female, and 31 were male. A total of 36 pneumothoraxes were small and were 34 large according to British Thoracic Society (BTS) criteria. We initially observed 56 patients with minimal or no symptoms (30 were large pneumothoraxes), with five requiring intercostal drainage (ICD). A total of 14 pneumothoraxes were treated with ICD as a first-line treatment: the mean age was 78 years (69–89) and eight patients had COPD. Five pneumothoraxes were large. All had significant symptoms. All pneumothoraxes resolved within six weeks with no associated mortality.

**Keywords:** pneumothorax; permanent pacemaker; cardiac-device implantation

## 1. Introduction

Cardiac devices that can be implanted to reduce morbidity and mortality include permanent pacemakers (PPM) and implantable cardioverter defibrillators. With rapid population expansion and increasing indications for device implantation, the number of implants has increased [1,2]. Any invasive procedure carries a risk of complications, and cardiac-device implantation is no different.

Often quoted complications are pneumothoraxes, infections, bleeding, myocardial perforation, and lead displacement, with overall complication rates being approximately four to five per cent [3–5]. Complications are often related to how vascular access was performed (for example subclavian or axillary vein or cephalic vein approaches) [5,6]. Specifically, pneumothorax during venous access is a complication of approximately 0.2–3.7% of procedures; however, some approaches, such as the cephalic vein approach, allow for a reduction in pneumothorax rates [7]. Pneumothorax risk seems to also be increased in female patients with concurrent chronic obstructive pulmonary disease, and is understandably associated with increased costs [3,4].

The management of iatrogenic pneumothorax is not well established. In the United Kingdom, the British Thoracic Society (BTS) guidelines are now over a decade old and only mention that most "will resolve with no intervention" [8]. A PubMed search with the terms "pneumothorax" OR "cardiac device implantation" OR "pacemaker" AND "management" revealed 24,050 results from December 1981 to December 2021 but none were directly related to guidance regarding post cardiac-device implantation pneumothoraxes. Two studies

revealed the use of needle aspiration and unidirectional valves in such pneumothoraxes in retrospective case series [9,10].

The evidence around pneumothorax management has evolved over time with some large randomised trials, the results of which have been recently published. Brown et al. [11] have shown that simple observation in primary spontaneous pneumothorax (pneumothorax occurring in patients with no discernible lung disease) can be feasible in a specific subset of patients. In the same vein, Gerhardy et al. [12] and Walker et al. [13] have shown that conservative management of traumatic pneumothoraxes is feasible and this concept will be tested in a randomised controlled trial in the United Kingdom [14]. Observation of selected secondary spontaneous pneumothoraxes is also feasible [15]. These studies will be further discussed later in this paper.

Northumbria Healthcare NHS Foundation Trust is a large district general hospital in the North East of England. We provide a well-established regional pleural service with advanced procedures, such as medical thoracoscopy, indwelling pleural catheter insertion, and more recently ambulatory pneumothorax and chest trauma pathways [16–18]. Our local approach has always been guided by local experience and the literature available at the time. The size of the pneumothorax appears to be less important than the symptoms. We have examined the outcomes of patients with pneumothorax post image-guided biopsy [19] and demonstrated, in an observational study, that those with large, asymptomatic pneumothoraxes post biopsy can be observed. Our local cardiology and respiratory services agreed that patients with pneumothoraxes post cardiac-device implantation should be assessed in a similar fashion for symptoms of dyspnoea and that pleural interventions be offered only if the symptoms are progressive.

The local incidence of pneumothorax post cardiac-device implantation is not known and an analysis of the subsequent management has not been previously performed.

## 2. Materials and Methods

### 2.1. Local Ethical Approval

Local Caldicott approval (Reference: RPI-C3737, obtained on 18 March 2021) was sought and granted from Northumbria HealthCare NHS Foundation Trust Information Governance Department, North Tyneside, United Kingdom for this retrospective study of cardiac-device implantations performed between 1 January 2010 and the 31 December 2020 from the local database kept by the cardiology department.

### 2.2. Aims of Study

The main aims of the study were to assess if those iatrogenic pneumothoraxes with no symptoms were indeed observed, how many went on to require intervention, and if those patients had any clinical conditions in common.

### 2.3. Inclusion Criteria and Exclusion Criteria

We logged all cardiac-device implantation devices onto the database. We then reviewed those patient episodes for their venous approach if relevant, in addition to any post-procedural complications. We excluded those without a post-procedure pneumothorax and further reviewed those identified as being complicated by a pneumothorax in the database. Then, we confirmed pneumothorax using contemporary radiology. We further analysed those patient records regarding basic demographics, pleural interventions required, and eventual or immediate outcomes collected from the notes. We collected long-term outcomes from primary care records if available.

### 2.4. Size Definition for Pneumothorax

We determined the size of pneumothorax by established BTS guidelines, where the differentiation of a "large" from a "small" pneumothorax is the presence of a visible rim of more than two centimetres (cm) between the lung margin and the chest wall. We digitally

measured the size on the hospital's PACS (picture archiving and communication system) system and independently verified it with a respiratory consultant.

*2.5. Analysis*

We applied a descriptive statistical methodology. We presented continuous data as median and interquartile ranges (IQRs) if outliers were present, or mean and ranges if normally distributed. We presented categorical variables as frequencies or percentages. We performed the analysis on Microsoft Excel 2019 (Microsoft 365).

## 3. Results and Discussion

A total of 2056 implantation episodes from January 2010 to December 2020 were reviewed.

Seventy pneumothoraxes (3.4%) were identified on post-implantation chest radiographs and clinical notes.

All were related to PPM insertion. A total of 59 (84.2%) of those PPM insertions were performed with a subclavian vein puncture approach and 1 (1.4%) had a cephalic vein puncture. The notes were incomplete in 10 (14.4%) patients, with no mention of the type of puncture employed.

The median age for those 70 patients with pneumothoraxes was 77.5 years (IQR 11). A total of 39 (56%) patients were female and 31 (44%) were male. All pneumothoraxes were on the side of the PPM (three on the right side, 67 on the left side). A total of 36 pneumothoraxes (51%) were small and 34 (49%) were large according to the above-described BTS criteria.

The above-described results are depicted in Table 1 below.

**Table 1.** Characteristics of study population: British Thoracic Society (BTS) criteria for pneumothorax categorisation: differentiation of a "large" from a "small" pneumothorax is the presence of a visible rim of more than two centimetres (cm) between the lung margin and the chest wall on a chest radiograph.

| Summary of Patient Characteristics, Pneumothoraces and Outcomes | | |
|---|---|---|
| Total number of pneumothoraces | 70 (3.4%) | All were post permanent pacemaker insertion |
| Approach for venous puncture (missing data in ten, 14%) | Subclavian 42 (60%) | Cephalic 18 (26%) |
| Median age for all pneumothoraces, in years | 77.5 | IQR 11 |
| Sex of patients with pneumothoraces | 31 (44%) male | 39 (56%) |
| Size of pneumothoraces (defined by British Thoracic Society criteria) | 36 (51%) small | 34 (59%) large |
| Immediate symptoms—all treated with chest drain | Present in 9 small pneumothoraces | Present in 5 large pneumothoraces |

None of the 70 patients developed hypoxaemia on peripheral oxygen saturation monitoring. Median peripheral oxygen saturations pre-procedure was 96% (IQR 3), and post-procedure it was 95% (IQR 2).

Fifty-six (80%) patients had no initial respiratory symptoms, such as chest pain or worsening breathlessness, from their baseline. A total of 30 (54%) of those were large pneumothoraxes, as defined by BTS criteria. Eighteen (32%) of those patients had concurrent chronic obstructive pulmonary disease. All 56 patients were observed initially with overnight admission. Five (9%) of those underwent intercostal drainage with a small-bore (12 French gauge) drain the next day due to the development of progressive breathlessness. All five of those had chronic obstructive pulmonary disease. Two of those five who underwent intercostal drainage initially had large pneumothoraxes and three were initially small. The median length of stay for those not requiring drainage was 2.1 days (IQR 2).

Fourteen (20%) patients with post-procedural pneumothoraxes were immediately treated with small-bore intercostal drains due to the development of severe symptomatic dyspnoea and/or chest pain. The median age in this particular group was 78 years (IQR 10, range 69–89). Eight of the patients had concurrent chronic obstructive pulmonary disease. Five (36%) of the pneumothoraxes were large at the outset. The median length of stay was 4.2 days (IQR 2).

All the pneumothoraxes were resolved on follow-up chest radiographs. There was no associated mortality. There were no incidents related to intercostal chest drain insertion.

Figure 1 below depicts the treatment pathway of those patients with pneumothoraxes post cardiac-device implantation according to their symptoms.

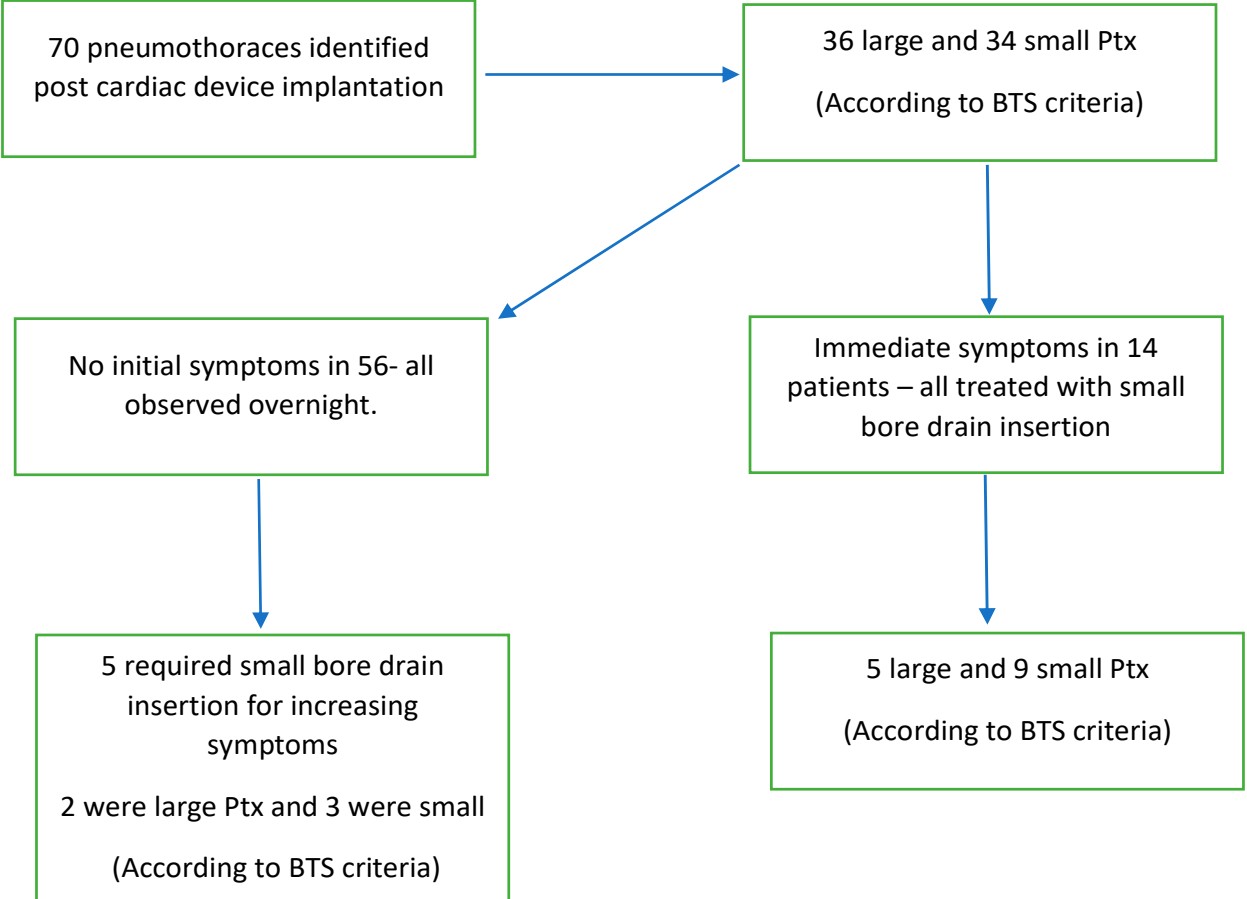

**Figure 1.** Treatment pathway of patients with pneumothoraxes (ptx) after cardiac-device implantation, according to size criteria.

1.   British Thoracic Society (BTS) criteria for pneumothorax size: differentiation of a "large" from a "small" pneumothorax is the presence of a visible rim of more than two centimetres (cm) between the lung margin and the chest wall on a chest radiograph;
2.   ICD: intercostal drain insertion;
3.   Discussion.

Our single-centre analysis of pneumothoraxes post cardiac-device implantation shows that in the absence of symptoms, or cardiovascular and respiratory decompensation, no pleural intervention is required, irrespective of the size of the pneumothorax. Patients with co-existing chronic obstructive pulmonary disease were more likely to have intercostal drainage after a pneumothorax but the number of patients was too small to infer any statistical significance. Specifically, in this study, the rate of pneumothorax post permanent-pacemaker insertion is within the known incidence percentage, which has a range of 0.2–3.7% [1–3].

### 3.1. Evidence towards Observation of Pneumothorax

The now-outdated 2010 BTS Pleural Disease guidelines suggest that for any pleural intervention, the choice of instrument (needle aspiration, small- or large-bore chest drain insertion and, by extension, any of the now-available ambulatory devices) should be

dictated by the expertise available locally [8]. Locally, the use of needle aspiration for any type of pneumothorax is not commonplace and anecdotally has been associated with an unacceptable failure of pneumothorax re-expansion, with the patients subsequently needing another procedure. Nevertheless, there is evidence from a very small case series that suggests that needle aspiration alone might be feasible in some instances [9]. This approach needs to be validated in large multi-centre trials before widespread applicability. Thus, we performed small-bore intercostal drain insertion only when an intervention was clearly required. This is reflected in the above findings that only intercostal drains were performed if an intervention was required.

The point about observations regarding primary spontaneous pneumothorax (pneumothorax occurring in patients with no apparent underlying lung disease) is made in the 2010 BTS guidance, where it is recommended that some large primary pneumothoraxes can be safely observed [8]. Observation for pneumothorax is not a new concept [20]. Stradling et al. in 1966 described a decade-long case series of 119 pneumothoraxes in 111 patients and approximately 80% of this unselected group of patients were managed conservatively with no apparent ill effects [20]. Prior to this, Kircher et al in 1957 described the rate of re-absorption of pleural air at 1.25–1.8% of the volume of pneumothorax every 24 h and argued to expect complete re-expansion at 7 weeks [21]. Expert opinion, which we would agree with, suggests that the patients and their symptoms should be treated rather than waiting for the appearance of the chest radiographs [22]. Walker et al. have recently eloquently written about intercostal drain insertion potentially exacerbating any air leak. The visceral pleural defect in any pneumothorax might be very self-limiting and intercostal drainage might well increase airflow across that defect; thus, the authors suggest that the avoidance of intervention may be desirable in minimally symptomatic patients [23]. As mentioned in the introduction, there are some newer clinical studies that have reported favourable outcomes for pneumothorax observation. Despite its limitations (long recruitment period, minimally symptomatic population, and lack of generalisable applicability), Brown et al. have shown that the conservative management of primary spontaneous pneumothorax was non-inferior to interventional management, with a lower risk of serious adverse events [11]. Only 15% of patients managed conservatively required subsequent chest drain insertion during the study period, and of these patients, only 2% were because of enlarging pneumothorax (other indications were pain and hypoxaemia). In secondary spontaneous pneumothorax, Gerhardy et al. showed that in 64 patients with secondary pneumothoraxes with a greater than one-centimetre pneumothorax, 39% of the managed patients did not require subsequent intervention, and their overall length of stay was shorter [15].

As also mentioned above in the introduction, there is no specific guidance describing the management of pneumothorax after cardiac-device implantation. An analogy might be made to traumatic pneumothoraxes, where small and/or asymptomatic pneumothoraxes can be observed, although formal randomised trial data needs to be generated [13,14]. Our centre has recently described our local experience [16].

One might argue that the insertion of a small-bore intercostal drain or performing a needle aspiration are simple and safe procedures; however, we would argue that this is only the case in experienced hands and that complications do exist. We have previously published our experience regarding intercostal chest drain and needle aspirations complications, and we described a 3% rate of drain "fall-out", a 0.5% rate of bleeding, and an approximately 4% rate of surgical emphysema [24]. These findings are not novel; for example, Sundaralingam et al. performed an elegant narrative review and found even higher rates of drain displacement at nearly 7% [25].

Our proposed management system is depicted in Figure 2 below:

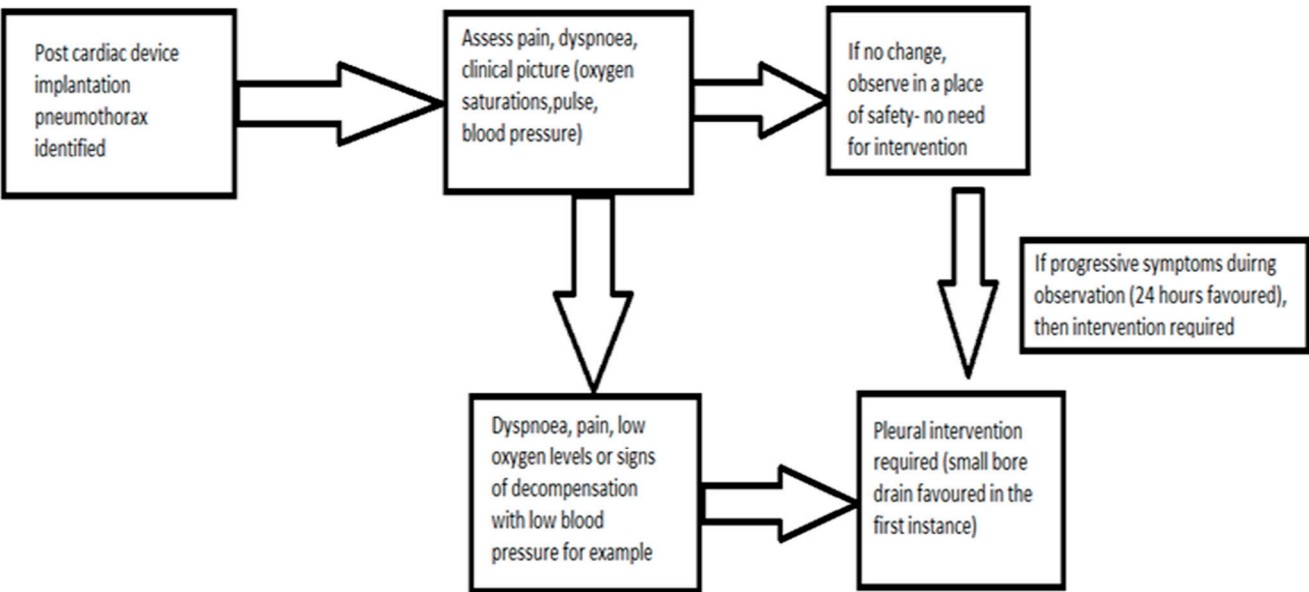

**Figure 2.** Suggested flow chart for assessment and management of pneumothorax post cardiac device implantation.

*3.2. Limitations*

The limitations of our study are manifold: it is a single-centre retrospective series with no control arm (a proposed controlled arm would a group without pneumothorax and pre-defined numbers to recruit to each arm for a powered statistical analyses). The initial recommendations for simple observation rather than treatment were based on expert opinion and on local experience rather than a strong evidence base. That evidence base is slowly coming together, as described above. We have successfully applied the same principles to patients with pneumothorax after image-guided biopsies and found that the vast majority of procedural pneumothoraxes can be observed and a symptom-based approach can be employed [18]. We suggest this as well in this specific patient group. However, all of the above informs the need for further studies in this field, and looking at iatrogenic pneumothoraxes in general. Due to the small numbers of patients, no meaningful statistical analysis was possible.

Furthermore, we do not differentiate between different devices and approaches within the pacemaker-using population (for example a single/dual chamber pacemaker may require full cephalic vein access) versus the population using cardiac resynchronization therapy (CRT) or an implantable cardioverter defibrillator (ICD), which may potentially be associated with severe complications, including pneumo- and haemothorax. As Vogler et al. [26]. suggest, different approaches, such as a triple-lead cephalic versus subclavian vein approach, can be feasible. Alternative techniques for left-ventricular pacing in cardiac resynchronization therapy may today include his pacing, as described by Senes et al. [27], or left bundle branch pacing, as described by Liu et al. [28]., leading to a potentially different vein approach. Furthermore, the age of patients presenting with complex arrhythmias is increasing (clearly demonstrated by Fumagalli et al. [29]), and this frailty may impact on cardiac-device implantation through a variety of processes. We also did not collect data on the "frailty syndrome", an emerging clinical problem in the everyday management of clinical arrhythmias. An ageing population with increasing incidences of renal failure, dementia, disability, atrial fibrillation, heart failure, falls, and cancer leads to elderly and frail individuals with enhanced susceptibility to stressors and a decreased capability for homeostasis [30].



## 4. Conclusions

Pneumothorax after cardiac-device implantation can occur in approximately 3% of patients. These pneumothoraxes can also be observed if patients are asymptomatic. There are some significant limitations to this retrospective study but this could pave the way for large, randomised, and controlled trials in iatrogenic pneumothoraxes.

**Author Contributions:** Conceptualization, A.A.; methodology, A.A.; software, A.A.; validation, A.A.; formal analysis, A.A.; investigation, G.G.; writing—original draft preparation, A.A.; writing—review and editing, A.A. and G.G. All authors have read and agreed to the published version of the manuscript.

**Funding:** This research received no external funding.

**Institutional Review Board Statement:** The study was conducted in accordance with the Declaration of Helsinki, and approved by the Institutional Review Board (Information Governance Department) of Northumbria Healthcare NHS Foundation Trust (protocol code RPI-C3737 and date of approval was 12 April 2021).

**Informed Consent Statement:** Patient consent was waived due to the study being an anonymized review.

**Data Availability Statement:** Anonymized data is available with reasonable requests.

**Acknowledgments:** We thank Honey Thomas for providing access to the cardiology database.

**Conflicts of Interest:** The authors declare no conflict of interest.

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
