# Peer review of "Cardiac-Device Implantation and Pneumothorax—A Symptom-Based Approach: Experience from a District General Hospital"

_reports, doi:10.3390/reports5040039_

Round 1

Reviewer 1 Report (New Reviewer)

I would congratulate for this very good paper documenting pneumothorax rates following cardiac device implantation as low; in particular in conclusions, irrespective of size, such iatrogenic pneumothoraces with no or minimal symptoms can often be observed with adequate safety. I completely agree according my clinical practice. Here you find comments in order to improved the manuscript

Please add a new paragraph entitled limitations. In particular:

1.In limitations authors should discuss an important topic (please cite 10.1038/s41598-018-35994-0): in particular, they did not differentiate a pacemaker-population (a single/dual chamber pacemakers may mostly require a full cefalic vein access) versus an ICD/CRT population potentially associated with severe complications including pneumo- and hemothorax (most DF4 implantations as well as CRT could more require a subclavian access ), although alternative techniques for left ventricular pacing in cardiac resynchronization therapy may today include His pacing (DOI: 10.1111/pace.14336) or Left Bundle Branch Pacing (DOI: 10.3389/fcvm.2021.630399) leading to a potentially different vein approach. Authors should clearly discuss these points in the “limitations” section, and should include all 4 fundamental suggested  references

2. In the limitations authors shoud also focus on the Frailty syndrome, an emerging clinical problem in the everyday management of clinical arrhythmias. In particular, the age of patients presenting with complex arrhythmias is increasing, and authors should discuss how the frailty may impact on a cardiac device implantation, potentially leading to a proper vein access. Plase amplify the discussion and cite fundamental refereces: 1)  DOI: 10.1007/s40520-018-1088-5 ; 2) https://doi.org/10.1093/europace/eux288

3. A nice figure explaining author’s proposed management /approach of patients with pneumothorax following cardiac device, is definitely welcome for every reader

4. Table 1 is not clear and also I would amplify the study population characteristics. Please resubmit in a perfect version

Author Response

I would congratulate for this very good paper documenting pneumothorax rates following cardiac device implantation as low; in particular in conclusions, irrespective of size, such iatrogenic pneumothoraces with no or minimal symptoms can often be observed with adequate safety. I completely agree according my clinical practice. Here you find comments in order to improved the manuscript

Please add a new paragraph entitled limitations. In particular:

1.In limitations authors should discuss an important topic (please cite 10.1038/s41598-018-35994-0): in particular, they did not differentiate a pacemaker-population (a single/dual chamber pacemakers may mostly require a full cefalic vein access) versus an ICD/CRT population potentially associated with severe complications including pneumo- and hemothorax (most DF4 implantations as well as CRT could more require a subclavian access ), although alternative techniques for left ventricular pacing in cardiac resynchronization therapy may today include His pacing (DOI: 10.1111/pace.14336) or Left Bundle Branch Pacing (DOI: 10.3389/fcvm.2021.630399) leading to a potentially different vein approach. Authors should clearly discuss these points in the “limitations” section, and should include all 4 fundamental suggested  references

This is an important point, and we find that quite a few of these articles are beyond our scope for a full discussion as we are respiratory physicians. Nevertheless. We have mentioned all of them and made our limitation section more visible. There are only 3 references in this part though, and we have referenced all of them

  1. In the limitations authors shoud also focus on the Frailty syndrome, an emerging clinical problem in the everyday management of clinical arrhythmias. In particular, the age of patients presenting with complex arrhythmias is increasing, and authors should discuss how the frailty may impact on a cardiac device implantation, potentially leading to a proper vein access. Plase amplify the discussion and cite fundamental refereces: 1) DOI: 10.1007/s40520-018-1088-5 ; 2) https://doi.org/10.1093/europace/eux288

Thank you and this is an important point. We have referenced both of those articles and elaborated on them.

  1. A nice figure explaining author’s proposed management /approach of patients with pneumothorax following cardiac device, is definitely welcome for every reader

This has been done, thank you

  1. Table 1 is not clear and also I would amplify the study population characteristics. Please resubmit in a perfect version

This has been reformatted, thank you

Reviewer 2 Report (New Reviewer)

This is overall a well-written manuscript. The authors put great effort into this work. It can be accepted at the current version.

Author Response

Thank you for your comments. No further action is required. 

Reviewer 3 Report (New Reviewer)

In this article, the authors aim to analyze the relationship between cardiac device implantation and pneumothorax through retrospective analysis. 

The abstract section should be revised. First, following the authors' guidelines, "the abstract should be a total of about 200 words maximum. The abstract should be a single paragraph and should follow the style of structured abstracts, but without headings..." (https://www.mdpi.com/journal/reports/instructions). moreover, after a general background, the authors should clearly describe the article's aims.

The introduction section should be improved. Lines 50-54: the authors stated, "A PubMed search with the terms pneumothorax OR cardiac device implantation OR pacemaker AND management revealed 20.050 results, but none directly related to guidance of post-cardiac device implantation pneumothorax.", without indicating the period included articles (please insert the start/stop dates). Lines 76-77: the authors should clearly describe the study's aims. Moreover, the use of term "review" should be checked: in a scientific article, it is commonly used referring to a literature review, while the authors mean a retrospective study. Please, check it.  

The "material and methods" should be extensively improved. First, I suggest subdividing it into subsections. Moreover, I suggest inserting the start/stop dates of all included cases, and inclusion and exclusion criteria.

The results section should be improved. For example, it could be of interest to analyze the year subdivision: are these cases more frequently recently? Moreover, several important data are missed (treatment, health status of each patient, drug treatment,): if these data are not available, please insert several considerations in the discussion sections, including them as limitations.

The "discussion" and "conclusion" sections should be improved. The authors should arrange the discussion, improving the take-home message. Moreover, more recent references should be inserted. Finally, revise the limitations section and the conclusion.

Minor points:

- the use of a short form should be preceded by an extension form (i.e., British Thoracic Society (BTS). Please, check all text.

- line 153: are the numbers references? If so, please check the author's guidelines.

Author Response

In this article, the authors aim to analyze the relationship between cardiac device implantation and pneumothorax through retrospective analysis. 

The abstract section should be revised. First, following the authors' guidelines, "the abstract should be a total of about 200 words maximum. The abstract should be a single paragraph and should follow the style of structured abstracts, but without headings..." (https://www.mdpi.com/journal/reports/instructions). moreover, after a general background, the authors should clearly describe the article's aims.

Thank you for this. I have revised the abstract accordingly and it is now 199 words, and the aims are clearly stated.

The introduction section should be improved. Lines 50-54: the authors stated, "A PubMed search with the terms pneumothorax OR cardiac device implantation OR pacemaker AND management revealed 20.050 results, but none directly related to guidance of post-cardiac device implantation pneumothorax.", without indicating the period included articles (please insert the start/stop dates). Lines 76-77: the authors should clearly describe the study's aims. Moreover, the use of term "review" should be checked: in a scientific article, it is commonly used referring to a literature review, while the authors mean a retrospective study. Please, check it.  

Thank you for those comments. We have added the dates: it was from 1981 to 2021. The main aims of the study were to assess if those iatrogenic pneumothoraces with no symptoms were indeed observed, and how many went on to require an intervention and if those patients had any clinical conditions in common. This is expanded upon more in the material and methods section. We have removed the word review and changed the sentence in line 69-70

The "material and methods" should be extensively improved. First, I suggest subdividing it into subsections. Moreover, I suggest inserting the start/stop dates of all included cases, and inclusion and exclusion criteria.

Thank you for those comments. We have made subsections and added the dates in line 74.

The results section should be improved. For example, it could be of interest to analyze the year subdivision: are these cases more frequently recently? Moreover, several important data are missed (treatment, health status of each patient, drug treatment,): if these data are not available, please insert several considerations in the discussion sections, including them as limitations.

Thank you for these comments. Unfortunately, as you mention, we did not collect this data and we did not collect the dates as well of the pneumothoraces. We have expanded the limitation section quite considerably and thinFk that perhaps the next iteration of this paper would definitely require such data to be collected.

The "discussion" and "conclusion" sections should be improved. The authors should arrange the discussion, improving the take-home message. Moreover, more recent references should be inserted. Finally, revise the limitations section and the conclusion.

 We believe that by stating the aims earlier on, the discussion section reads better now. We have also added a subsection and I am afraid that there are no real new references regarding the observation of iatrogenic pneumothoraces. The most interesting paper is by Walker et al and they still reference the old but quite relevant papers. However, we have added quite a few newer references. We have added a proposed management system as well in this section. The limitations section has been greatly improved as well.

Minor points:

- the use of a short form should be preceded by an extension form (i.e., British Thoracic Society (BTS). Please, check all text. This has been done

- line 153: are the numbers references? If so, please check the author's guidelines.yes they are, we have revised them/

Round 2

Reviewer 1 Report (New Reviewer)

Congratulation for the paper. Authors carefully followed all suggestions, manuscript is now ready since it definitely improved

Author Response

Thank you- no further changes are required

Reviewer 3 Report (New Reviewer)

Following the reviewers' suggestions, the authors have improved their manuscript. I endorse the publication in its current form.

Author Response

Thank you for the comments. No further action is required

This manuscript is a resubmission of an earlier submission. The following is a list of the peer review reports and author responses from that submission.

Round 1

Reviewer 1 Report

The main problem of this article is having no novelty for the treatment strategy of pneumothorax, also giving no effective reference for the future practice.

The treatment indications have already been standardized.

I suggest that this article should better be written with the aim to find out what caused the pneumothorax  and how to prevent. That surely will help and reduce the morbidity.

Reviewer 2 Report

the manuscript cannot be improved